# Humans prioritize walking efficiency or walking stability based on environmental risk

**Ashwini Kulkarni**[ID], **Chuyi Cui**, **Shirley Rietdyk**, **Satyajit Ambike**[ID]*

Department of Health and Kinesiology, Purdue University, West Lafayette, Indiana, United States of America

 These authors contributed equally to this work.
* sambike@purdue.edu

**Data Availability Statement:** All data and code are available on the data repository hosted by Purdue University. Kulkarni, A.; Chuyi Cui; Rietdyk, S.; Ambike, S. S. (2022). Data and Code from: Humans prioritize walking efficiency or walking

## Abstract

In human gait, the body's mechanical energy at the end of one step is reused to achieve forward progression during the subsequent step, thereby reducing the required muscle work. During the single stance phase, humans rely on the largely uncontrolled passive inverted pendular motion of the body to perpetuate forward motion. These passive body dynamics, while improving walking efficiency, also indicate lower passive dynamic stability in the anterior direction, since the individual will be less able to withstand a forward external perturbation. Here we test the novel hypothesis that humans manipulate passive anterior-posterior (AP) stability via active selection of step length to either achieve energy-efficient gait or to improve stability when it is threatened. We computed the AP margin of stability, which quantifies the passive dynamic stability of gait, for multiple steps as healthy young adults (N = 20) walked on a clear and on an obstructed walkway. Participants used passive dynamics to achieve energy-efficient gait for all but one step; when crossing the obstacle with the leading limb, AP margin of stability was increased. This increase indicated caution to offset the greater risk of falling after a potential trip. Furthermore, AP margin of stability increased while approaching the obstacle, indicating that humans proactively manipulate the passive dynamics to meet the demands of the locomotor task. Finally, the step length and the center of mass motion co-varied to maintain the AP margin of stability for all steps in both tasks at the specific values for each step. We conclude that humans actively regulate step length to maintain specific levels of passive dynamic stability for each step during unobstructed and obstructed gait.

## 1. Introduction

Gymnasts exploit the mechanical properties of their bodies to achieve remarkable feats. While performing a triple axel, for example, an ice-skater holds her arms close to or farther from her body to modulate her angular velocity. High velocity during the flight phase allows for more turns [1], whereas lower velocity close to landing improves safety [2].

The ability and propensity to adapt the body's mechanical properties to facilitate motion is not limited to trained sportspersons; it is a feature of the walking patterns of most adults. On clear and level walkways, adults use a characteristic step length to recycle the kinetic energy at

stability based on environmental risk. Purdue University Research Repository. doi:10.4231/J887-SN98.

**Funding:** AK: No Grant Number. Bilsland Dissertation Fellowship, Purdue University. https://www.purdue.edu/gradschool/fellowship/funding-resources-for-students/fellowships/managed-fellowships/bilsland-dissertation-fellowship.html CC: No grant number. Purdue University Department of Health and Kinesiology, Templin Graduate Student research award. https://www.purdue.edu/hhs/hk/graduate/scholarships/grants.html#:~:text=The%20Templin%20Graduate%20Student%20Research,research%20and%2For%20travel%20activities. The funders had no role in study design, data collection and analysis, decision to publish, or preparation of the manuscript.

**Competing interests:** The authors have declared that no competing interests exist.

the end of a step to rotate the body over the new stance foot. Part of the forward progression of the body is thus achieved passively, which improves walking efficiency [3,4]. However, if the environment poses a threat to stability, the passive forward motion can be altered to improve gait stability rather than efficiency. Such adaptation is suggested by the difference in the behaviors of young and older adults. While stepping over an obstacle, older adults rely less on the passive forward motion compared to young adults [5], since age-related declines in strength and coordination make it more difficult to recover from a trip. In other words, since, after a trip, forward passive motion makes a forward fall more likely, older adults proactively improve their passive stability by reducing this motion while crossing obstacles.

This tradeoff between stability and efficiency has been supported by the study of the margin of stability in the anterior-posterior direction ($MOS_{AP}$; abbreviations in Table 1). $MOS_{AP}$ has been used to identify differences in the recruitment of passive dynamics across age groups [5–9], in pathological populations [10–12], and in dual tasks [13]. $MOS_{AP}$, based on the inverted pendulum model of gait, and usually computed at heel contact, is the distance from the anterior boundary of the base of support (BOS; the leading heel) to the extrapolated center of mass (XcoM), which reflects the center of mass (CoM) state [14,15]. The condition XcoM ahead of the BOS boundary at heel contact indicates efficient gait: the body has sufficient kinetic energy to passively rock over the new stance ankle. More anterior location of XcoM indicates greater forward passive motion and greater efficiency. At the same time, a forward perturbation during the swing phase would accentuate the greater forward passive motion, making a forward fall more likely. Therefore, a more anterior location of the XcoM at heel contact is also interpreted as lower passive dynamic stability in the anterior direction [5,13].

Here, we advance in three ways our understanding of the role of passive dynamics during gait based on $MOS_{AP}$. First, we demonstrate that adaptation in passive dynamics in response to environmental threats to stability is apparent in young adults as well. We show that young adults increase passive dynamic stability while crossing an obstacle compared to unobstructed walking. Second, we demonstrate that the shift in passive dynamic stability occurs in the steps leading up to the obstacle. This novel finding is consistent with the notion that motor behavior usually does not change instantaneously, but evolves over characteristic times determined in part by the inertia of the body [16]. Third, we demonstrate that humans modulate step length

**Table 1. Abbreviations.**

| | |
|---|---|
| Anterior-posterior | AP |
| Medial-lateral | ML |
| Foot placements | fp |
| Center of mass | CoM |
| Center of mass velocity | $V_{CoM}$ |
| Leg length | $l$ |
| Acceleration due to gravity | $g$ |
| Base of support | BOS |
| Extrapolated center of mass | XcoM |
| Margin of stability in anterior-posterior direction | $MOS_{AP}$ |
| Uncontrolled manifold | UCM |
| Orthogonal manifold | ORT |
| Synergy index | $\Delta V$ |
| Synergy index z transformed | $\Delta V_Z$ |
| Variance along the uncontrolled manifold | $V_{UCM}$ |
| Variance along the orthogonal manifold | $V_{ORT}$ |

to control $MOS_{AP}$. We show that $MOS_{AP}$ is similar for the steps of unobstructed gait, and it is maintained at different values for specific steps while approaching and crossing an obstacle. Hof [15] showed that a consistent $MOS_{AP}$ yields a stable walking cycle in a mathematical model, and hypothesized that humans would benefit from this strategy as well [17]. It has been suggested that the variability rather than the average value of MOS better represents passive dynamic stability of human gait [18]. In the anterior-posterior (AP) direction specifically, indirect evidence for this hypothesis is provided by the fact that persons with Multiple Sclerosis (MS) exhibit greater $MOS_{AP}$ variability compared to healthy controls [11], and within the group of persons with MS, fallers show greater $MOS_{AP}$ variability than non-fallers [10].

We sought direct evidence for the control of $MOS_{AP}$ in the unobstructed and obstructed gait of healthy young adults. Since the $MOS_{AP}$ is a function of step length and XcoM, we hypothesized that the central nervous system responds to changes the XcoM with a corresponding correction in step length so that the $MOS_{AP}$ itself is relatively invariant at each heel contact. However, the specific values of $MOS_{AP}$ that are maintained via the covariation in XcoM and step length are different at various steps during obstructed gait.

We employed the uncontrolled manifold (UCM) method [19] to test this hypothesis. For each heel contact, we computed the synergy index that quantifies the covariation between XcoM and step length. A positive synergy index indicates that $MOS_{AP}$ was actively controlled, i.e., stabilized at the specific across-trial mean $MOS_{AP}$ for that step. A higher value indicates a stronger synergy or higher stability of $MOS_{AP}$. We emphasize that stability of $MOS_{AP}$ is different from the stability of gait. $MOS_{AP}$ is the measure of *passive* dynamic gait stability. In contrast, the synergy index indicates the efficacy of *active* control at stabilizing or maintaining $MOS_{AP}$ at a specific value for a given step.

We computed $MOS_{AP}$ and the synergy index for multiple steps for unobstructed gait trials and for trials where participants approached an obstacle, stepped over it, and continued walking till the end of the walkway. We hypothesized a task by step interaction for $MOS_{AP}$ (H1). The $MOS_{AP}$ will not be different across tasks (obstructed and unobstructed) for the steps at the start and end of the walkway, but the $MOS_{AP}$ will be higher for the approach steps 1–2 steps before the obstacle and for the crossing steps. These changes will indicate prioritization of safety over efficiency [5,6,20]. Next, we hypothesized that the synergy index will be greater than zero, indicating that the step length and the XcoM co-vary to stabilize $MOS_{AP}$ for all steps in both tasks (H2). Finally, we hypothesized a task by step interaction for the synergy index (H3). The synergy index will not be different across tasks for the steps at the start and end of the walkway, but the synergy index will be lower while crossing an obstacle placed in the middle of the walkway. The lower synergy index will reflect the greater motor demands associated with the crossing steps; larger muscle activations required for stepping over the obstacle [21] will be associated with greater noise [22], which will make stabilization more difficult.

## 2. Materials and methods

### 2.1 Participants

Twenty-six healthy young adults participated in the study. We excluded six participants due to poor kinematic tracking. Data from 20 participants (14 women, 22.3 ± 3.7 years, 1.7 ± 0.1 m, 66.9 ± 14.6 kg) were used for analysis. All participants walked without aid, had no orthopedic, neuromuscular, or dementia disorders, and were independent in daily activities. Vision was normal or corrected-to-normal. The study was approved by Purdue University's Institutional Review Board, and all participants provided written informed consent (Protocol number: IRB-2021-331).

## 2.2 Equipment and procedures

We assessed leg dominance using the Waterloo Footedness Questionnaire—Revised [23]. Participants walked at their self-selected speed on a 6.0 m walkway and stepped over an obstacle when present (Fig 1A). The obstacle was 100 cm wide × 0.4 cm deep. The height of the obstacle was scaled to 25% of the participant's leg length. The obstacle was made of black Masonite and designed to tip if contacted. The starting position was determined for each participant such that they took five steps before reaching the obstacle, crossed the obstacle naturally with the right leg first and stopped three to four steps later (Fig 1A).

Participants first performed 20 trials without an obstacle (no obstacle task), followed by 20 trials of walking with an obstacle (obstacle-crossing task). We collected kinematic data at 100 Hz with a motion capture system (Vicon Vero, Oxford, UK) with marker clusters placed bilaterally on the lower back, thigh, shank, and foot. We digitized the joint centers and posterior aspect of the heels to identify their locations relative to the marker clusters. We also digitized the top edge of the obstacle to identify its position.

## 2.3 Analysis

Across all trials and all participants, the obstacle was contacted and tipped 12 times out of 3360 trials (0.4% of trials). We did not include the contact trials in our analysis. Furthermore, some trials were discarded due to poor kinematic tracking. To have the same number of trials for all participants and tasks, we selected 15 trials with good kinematic data. Fifteen trials are sufficient for reliable quantification of the synergy variables [24]. We filtered all kinematic data using a zero-lag, $4^{th}$ order, low-pass Butterworth filter with a cut-off of 7 Hz. We identified seven foot placements (Fig 1A) using the AP position of the heel [25]. We quantified spatiotemporal gait parameters and margin of stability at heel contact at the seven foot placements ($fp_{-4}$ to $fp_{+3}$) and six steps ($Step_{-3}$ to $Step_{+2}$; Fig 1A). Step length was defined as the distance between two consecutive heel contacts. CoM position was computed as the centroid of the triangle formed by the left and right anterior superior iliac spines and the center of the left and right posterior superior iliac spines [26]. CoM velocity was obtained by differentiating the CoM position data. The extrapolated center of mass (XcoM) was calculated in the sagittal plane as [14]:

$$XcoM = CoM + \frac{V_{CoM}}{\sqrt{\frac{g}{l}}},$$

where CoM is the anterior-posterior CoM position, $V_{CoM}$ is the anterior-posterior CoM velocity, $g$ is the acceleration due to gravity, and $l$ is the participant's leg length (Fig 1B). Leg length was calculated as the sagittal-plane distance between the CoM and the ankle of the limb that contacted the ground. We used the average of the leg length values obtained from the 15 trials for each step to compute the XcoM for that step [18]. We computed $MOS_{AP}$ at the instant of leading heel contact in a coordinate frame fixed at the location of the rear heel contact (Fig 1B):

$$MOS_{AP} = Step\ length - XcoM, \tag{1}$$

i.e., the distance from the anterior boundary of the BOS, defined by the position of the leading heel, to the XcoM. Negative $MOS_{AP}$ (XcoM ahead of the anterior BOS boundary) at heel contact indicates that the body possesses sufficient energy to passively rotate beyond the upright position and fall forward (assuming that the energy loss at heel contact and the energy input during the subsequent push-off are either equal or negligible). Conversely, positive $MOS_{AP}$ (XcoM is behind the anterior BOS boundary) indicates that the body cannot passively rotate beyond the upright position, and it will eventually fall backward.

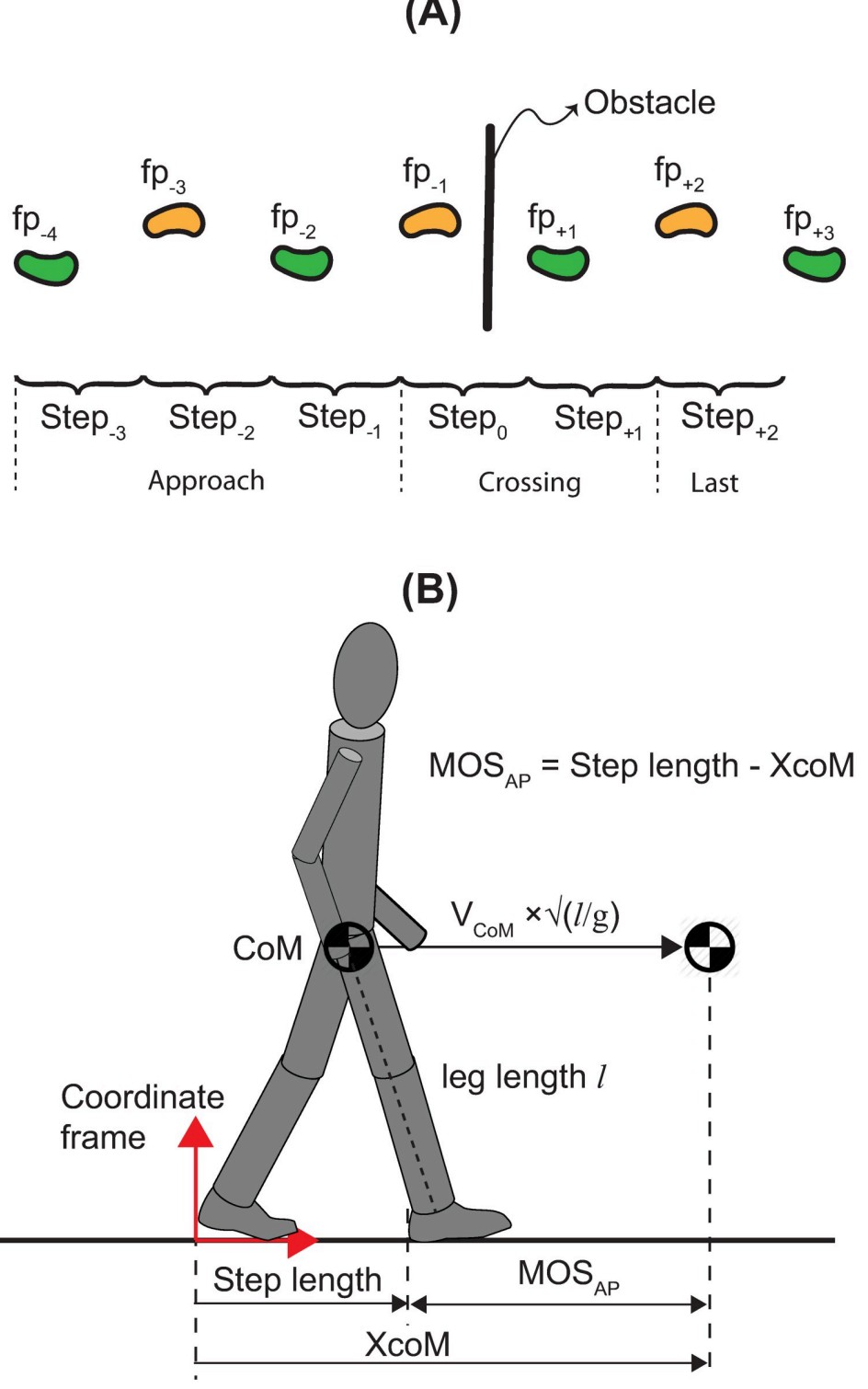

**Fig 1. Experimental task and basic definitions.** Illustration of foot placements while approaching (fp$_{-4}$ to fp$_{-1}$), crossing (fp$_{+1}$ and fp$_{+2}$) and after crossing (fp$_{+3}$) the obstacle, and of steps while approaching (Step$_{-3}$ to Step$_{-1}$), crossing (Step$_0$ and Step$_{+1}$) and after crossing (Step$_{+2}$) the obstacle **(A)**. Definitions of the extrapolated center of mass (XcoM) and margin of stability, MOS$_{AP}$. Step length, XcoM and MOS$_{AP}$ are computed at the moment of lead heel contact in a coordinate frame located where the rear heel contacted the ground **(B)**.

For a passive walker, $MOS_{AP} = 0$ serves as a threshold for detecting unstable gait. Hof demonstrated that stable gait arises with XcoM ahead of the BOS ($MOS_{AP} < 0$; Eq 1) in a mathematical model of a passive walker [15]. Conversely, $MOS_{AP} > 0$ would mean that the forward progression of the passive walker will stop, and the gait will be unstable.

In this work, we are interested in the forward loss of stability arising from a trip. Therefore, we invert the interpretation of the $MOS_{AP}$ from that for the passive walker. We consider $MOS_{AP}$ directly proportional to gait stability in the anterior direction. That is, lower $MOS_{AP}$ (more anterior location of the XcoM) indicates less stable gait, since a forward fall is more likely if a perturbation like a trip occurs. Conversely, higher $MOS_{AP}$ (more posterior location of XcoM) indicates more stable gait [13,27]. Furthermore, $MOS_{AP}$ is inversely proportional to efficiency: lower $MOS_{AP}$ indicates greater passive forward motion, and hence less active propulsion is required to maintain forward progression, and vice-versa. In this way, $MOS_{AP}$ reflects the tradeoff between stability and efficiency.

We used the uncontrolled manifold (UCM) analysis to quantify the synergy stabilizing $MOS_{AP}$ at heel contact. A *synergy* is co-variation in a redundant sets of input body variables that maintains important output variables that define task performance. The UCM method has been widely used to quantify the synergistic covariation in body variables in a variety of human movements including gait ([28,29]; see [30,31] for recent reviews). Importantly, in addition to identifying task-specific covariation, the UCM method identifies the salient task variables controlled by the nervous system.

Here, we use the UCM method to evaluate the hypothesis that the input variables–step length and XcoM, co-vary to stabilize the output variable, i.e., $MOS_{AP}$. We performed the analysis separately for each step for both tasks. We first obtain from the constraint equation (Eq 1), the Jacobian matrix that relates small changes in the step length and XcoM to changes in $MOS_{AP}$: J = [1–1]. The one-dimensional null space of this Jacobian defines the UCM, and its one-dimensional compliment defines the orthogonal (ORT) manifold. We pool the across-trial step length and XcoM data for a particular step. The deviation in the step length and XcoM data for each trial from the across-trial mean is projected onto the UCM and the ORT manifolds. The variances in these projections are the $V_{UCM}$ and the $V_{ORT}$, respectively. These variance components yield the synergy index:

$$\Delta V = \frac{V_{UCM} - V_{ORT}}{\left(\frac{V_{UCM} + V_{ORT}}{2}\right)}.$$

The synergy index $\Delta V$ has a threshold value of zero. When $\Delta V > 0$, $V_{UCM} > V_{ORT}$. In general, this implies that the control is organized so that most of the variability in the inputs is channeled along the UCM, and therefore, it does not alter task performance, i.e., the output. Here, $\Delta V > 0$ implies that the step length and XcoM covary to stabilize $MOS_{AP}$. Conversely, when $\Delta V < 0$, $V_{UCM} < V_{ORT}$. This implies that control is organized so that most of the variability in the inputs alters task performance (i.e., change $MOS_{AP}$). In both these cases, when the synergy index differs from zero, we would conclude that $MOS_{AP}$ is a controlled variable. Finally, $\Delta V = 0$ indicates that there is no task-specific co-variation in the step length and XcoM. This result would indicate that $MOS_{AP}$ is not a controlled variable.

The synergy index $\Delta V$ ranges from -2 to 2. Therefore, it was z-transformed for statistical analysis [29,32]:

$$\Delta Vz = \frac{1}{2} \times \log\left[\frac{2 + \Delta V}{2 - \Delta V}\right].$$

Note that $\Delta V = 0$ translates to $\Delta Vz = 0$. We report $\Delta Vz$ values in the Results section and use $\Delta Vz$ values to draw inferences, consistent with most previous studies [30].

## 2.4 Statistical analysis

To determine whether the gait task and foot placement affected $MOS_{AP}$ (H1), we performed a two-way (task × step) repeated measures ANOVA. To identify the source of changes in $MOS_{AP}$, we performed the two-way ANOVA separately on the CoM position, CoM velocity at heel contact and step length. To determine whether the synergies were present, we performed separate, one-sample t-tests to test if $\Delta Vz$ was significantly different from zero for each step in the two tasks (H2). To determine whether the gait task and foot placement affected the synergy index (H3), we performed a task × step repeated-measures ANOVA. To identify the source of changes in the synergy index, we performed the two-way ANOVA separately on the variance components ($V_{UCM}$, $V_{ORT}$). For all ANOVA tests, we fit a generalized linear model with random effects. Recall that the UCM analysis utilizes data from all 15 trials to yield a single value each for the synergy index, $V_{UCM}$, and $V_{ORT}$. Therefore, participant was the random effect for these ANOVAs. However, data from all 15 trials was used to analyze all the remaining variables. Therefore, the trial number within each participant was the nested random effect for these ANOVAs. Tukey-Kramer adjustments were used to perform the following planned pairwise comparisons: (1) across-task comparison at each of the six steps (or seven foot placements), and (2) all across-step (or foot placement) comparisons for the obstacle-crossing task only. All analyses were performed using the PROC GLIMMIX procedure in SAS 9.4 (Cary, NC, USA) with significance set at 0.05.

## 3. Results

The Waterloo Footedness Questionnaire scores were $8 \pm 5$, indicating that all participants were right-leg dominant.

Next, we use data from one participant (Fig 2) to outline the overall results. The detailed statistical results are presented in the following subsections. Fig 2 illustrates (1) the changes in various outcome variables while approaching, crossing, and resuming gait after crossing an obstacle, and (2) how the variables compare with unobstructed gait for each step. All outcome variables showed a task × step interaction (supporting H1 and H3) indicating that the pattern of changes in the variables across steps was different for the obstacle-crossing task compared to the no obstacle task. The largest across-task changes were observed in all outcomes for the two crossing steps ($Step_0$ and $Step_{+1}$), and across-task changes were apparent in some outcomes in the approach to the obstacle ($Step_{-3}$ to $Step_{-1}$). All variables showed across-step changes for the obstacle-crossing task. The synergy index was greater than zero, indicating that the step length and the XcoM co-varied to stabilize $MOS_{AP}$ for all steps in both tasks (supporting H2). In the detailed results below, we first present the results for $MOS_{AP}$, followed by results for the variables that constitute $MOS_{AP}$: CoM position relative to rear heel, CoM velocity at heel contact, and step length. We then present the results for the UCM outcome variables quantifying the stability of $MOS_{AP}$.

## 3.1 Margin of stability

**$MOS_{AP}$.** We observed a task × step interaction for $MOS_{AP}$ ($F_{6,4167} = 886.89$, $p < 0.001$; $\eta_p^2 = 0.9$; Fig 3A). Post-hoc comparisons across tasks revealed that $MOS_{AP}$ was higher (more stable) for the obstacle-crossing task than the no obstacle task at all except the first ($fp_{-4}$) and last two foot placements ($fp_{+2}$ and $fp_{+3}$). $MOS_{AP}$ was lower (less stable) for $fp_{+2}$ (after trail foot crossing) for the obstacle-crossing task. All pair-wise across-step comparisons for the obstacle-crossing task are depicted in Fig 3A. For brevity, we describe only some of the significant

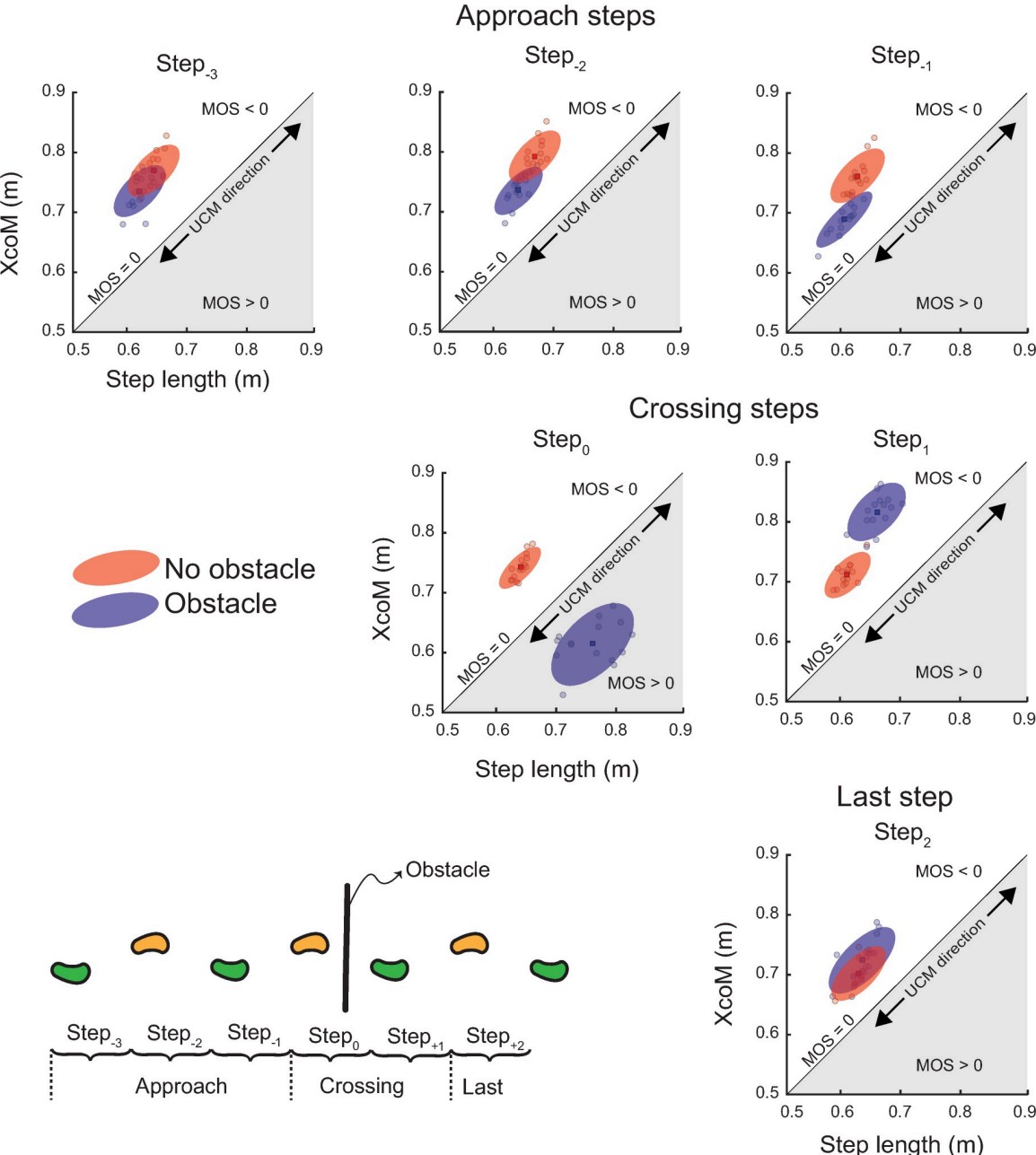

**Fig 2. Representative data from one participant.** Figure illustrates changes in various outcomes while approaching and crossing an obstacle, and then resuming unobstructed gait (blue ellipses). Steps during unobstructed gait are included for comparison (red ellipses). The 45˚ line separates the regions of positive and negative $MOS_{AP}$. The 45˚ line also represents the direction of the uncontrolled manifold (UCM). Variations in step length and XcoM along this direction do not change $MOS_{AP}$. The ellipses are centered at the centroid of the across-trial data for each step, and the major axes are aligned with the 45˚ line. However, the flatness of the ellipses is representative. Changes in mean values of the $MOS_{AP}$ are reflected in the position of the ellipses in each chart. The flatness of the ellipse reflects the synergy index, with flatter ellipses indicating higher synergy indices that reflect stronger covariation between the step length and XcoM.

differences. Changes in $MOS_{AP}$ occurred before the obstacle was reached; $MOS_{AP}$ was higher (more stable) for $fp_{-1}$ relative to the two preceding foot placements. For the two crossing steps, $MOS_{AP}$ first increased to the highest value for $fp_{+1}$, and then decreased to the lowest value for $fp_{+2}$ across all foot placements.

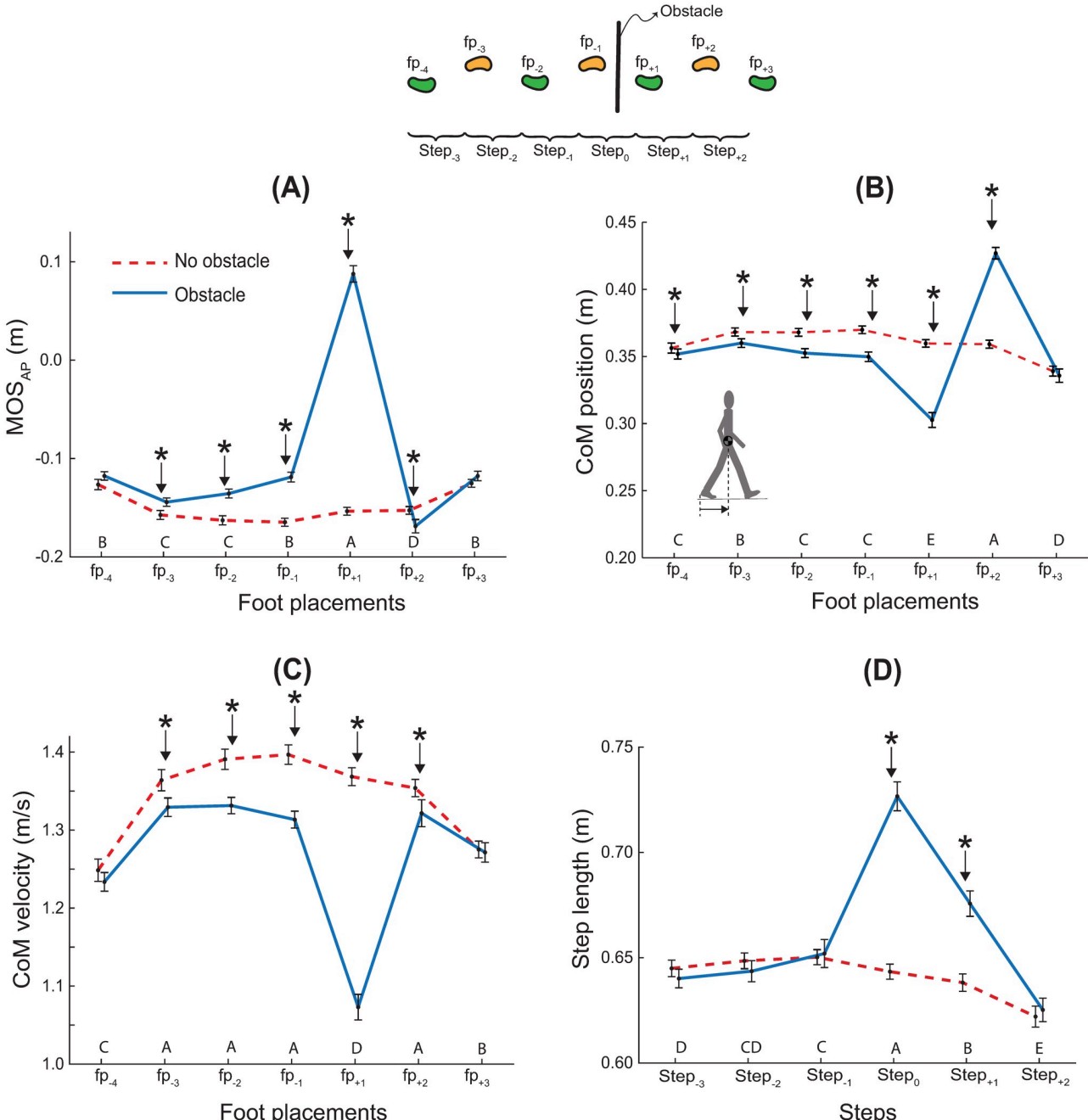

**Fig 3. MOS$_{AP}$ and its components.** MOS$_{AP}$ (**A**), CoM position at heel contact (**B**), CoM velocity at heel contact (**C**) and step length (**D**). Data are across-subject means (N = 20), and error bars denote standard error. Asterisks denote significant task difference at that step (p < 0.05). Across-step pairwise comparisons for the obstacle-crossing task are indicated by the row of letters above the horizontal axis for each panel. Steps that do not have letters in common are significantly different from each other (i.e., A is different from B, but A is not different from AB).

**CoM position relative to the rear heel.** A task × step interaction was observed for the CoM position relative to the rear heel ($F_{6,4167}$ = 391.20, p < 0.001; $\eta_p^2$ = 0.9; Fig 3B). Post-hoc comparisons across tasks revealed that CoM was closer to the rear heel at the foot placements of all approach steps and the lead crossing step (fp$_{-4}$ to fp$_{+1}$) for the obstacle-crossing

task than the no obstacle task. Conversely, CoM was closer to the front heel at the foot placement for the trail crossing step ($fp_{+2}$) for the obstacle-crossing task. All pair-wise across-step comparisons for the obstacle-crossing task are depicted in Fig 3B. For brevity, we describe only some of the significant differences. CoM position changed for the last three foot placements. The CoM was closer to the rear foot at the foot placement for the lead crossing step ($fp_{+1}$). It shifted forward, so that it was closer to the front heel at the foot placement for the trail crossing step ($fp_{+2}$). Finally, for the last step ($fp_{+3}$), CoM position was consistent with that for the approach steps.

**CoM velocity at heel contact.**   We observed a task × step interaction for the CoM velocity at heel contact ($F_{6,4167} = 161.52$, $p < 0.001$; $\eta_p^2 = 0.8$; Fig 3C). Post-hoc comparisons across tasks revealed that the CoM velocity was lower for the obstacle-crossing task compared to the no obstacle task at all but the first and the last foot placements ($fp_{-4}$ and $fp_{+3}$). Post-hoc comparisons across steps for the obstacle-crossing task revealed that the CoM velocity was lower at the first ($fp_{-4}$) and last ($fp_{+3}$) foot placement compared to other foot placements, except at the lead foot crossing ($fp_{+1}$), where the CoM velocity was lower compared to all other foot placements.

**Step length.**   We observed a task × step interaction for the step length ($F_{5,3569} = 149.09$; $p < 0.001$; $\eta_p^2 = 0.9$; Fig 3D). Post-hoc comparisons across tasks revealed that step length was longer during the obstacle-crossing task compared to the no obstacle task at the lead ($step_0$) and trail crossing ($step_{+1}$) steps. Post-hoc comparisons across steps for the obstacle-crossing task revealed that step length was higher for step-1 compared to step-3. That is, relative to the early step, the step length increased one step before the first crossing step. Step length was highest for the lead crossing step ($step_0$), and then it shortened for the trail crossing step ($step_{+1}$), but remained longer than all other steps. Finally, step length was the shortest for the last step ($step_{+2}$) compared to all other steps.

## 3.2 UCM variables

**Synergy index.**   The synergy index ($\Delta Vz$) was significantly greater than zero at all steps for both tasks ($t_{19} < 15.97$, $p < 0.01$; Fig 4A).

There was a task × step interaction for the synergy index $\Delta Vz$ ($F_{5,209} = 2.89$, $p = 0.015$; $\eta_p^2 = 0.3$; Fig 4A). Post-hoc comparisons across tasks revealed that $\Delta Vz$ was lower during the obstacle-crossing task compared to the no obstacle task for only the lead ($step_0$) and trail crossing steps ($step_{+1}$). Post-hoc comparisons across steps for the obstacle-crossing task revealed that $\Delta Vz$ was lower at the lead crossing step ($step_0$) compared to all non-crossing steps except $step_{-2}$, and $\Delta Vz$ was lower at the trail crossing step ($step_{+1}$) compared to the last step ($step_{+2}$).

**Variance components.**   There was a task × step interaction for $V_{UCM}$ ($F_{5,209} = 3.12$, $p = 0.009$; $\eta_p^2 = 0.3$; Fig 4B). Post-hoc comparisons across tasks revealed that $V_{UCM}$ was higher for the obstacle-crossing task compared to the no obstacle task at all but the first two steps ($step_{-3}$ and $step_{-2}$). Post-hoc comparisons across steps for the obstacle-crossing task revealed that $V_{UCM}$ was higher at the lead crossing step ($step_0$) compared to the first two steps ($step_{-3}$ and $step_{-2}$).

There was a task × step interaction for $V_{ORT}$ ($F_{5,209} = 16.51$, $p < 0.001$; $\eta_p^2 = 0.3$; Fig 4C). Post-hoc comparisons across task revealed that $V_{ORT}$ was higher during the obstacle-crossing task compared to the no obstacle task at the lead ($step_0$) and trail ($step_{+1}$) crossing steps. Post-hoc comparisons across steps for the obstacle-crossing task revealed that $V_{ORT}$ was higher at the lead crossing step ($step_0$) compared to all other steps, and $V_{ORT}$ was higher at the trail crossing step ($step_{+1}$) compared to all steps except the lead crossing step (lower for $step_{+1}$ compared to $step_0$) and the preceding step ($step_{-1}$; no significant difference).

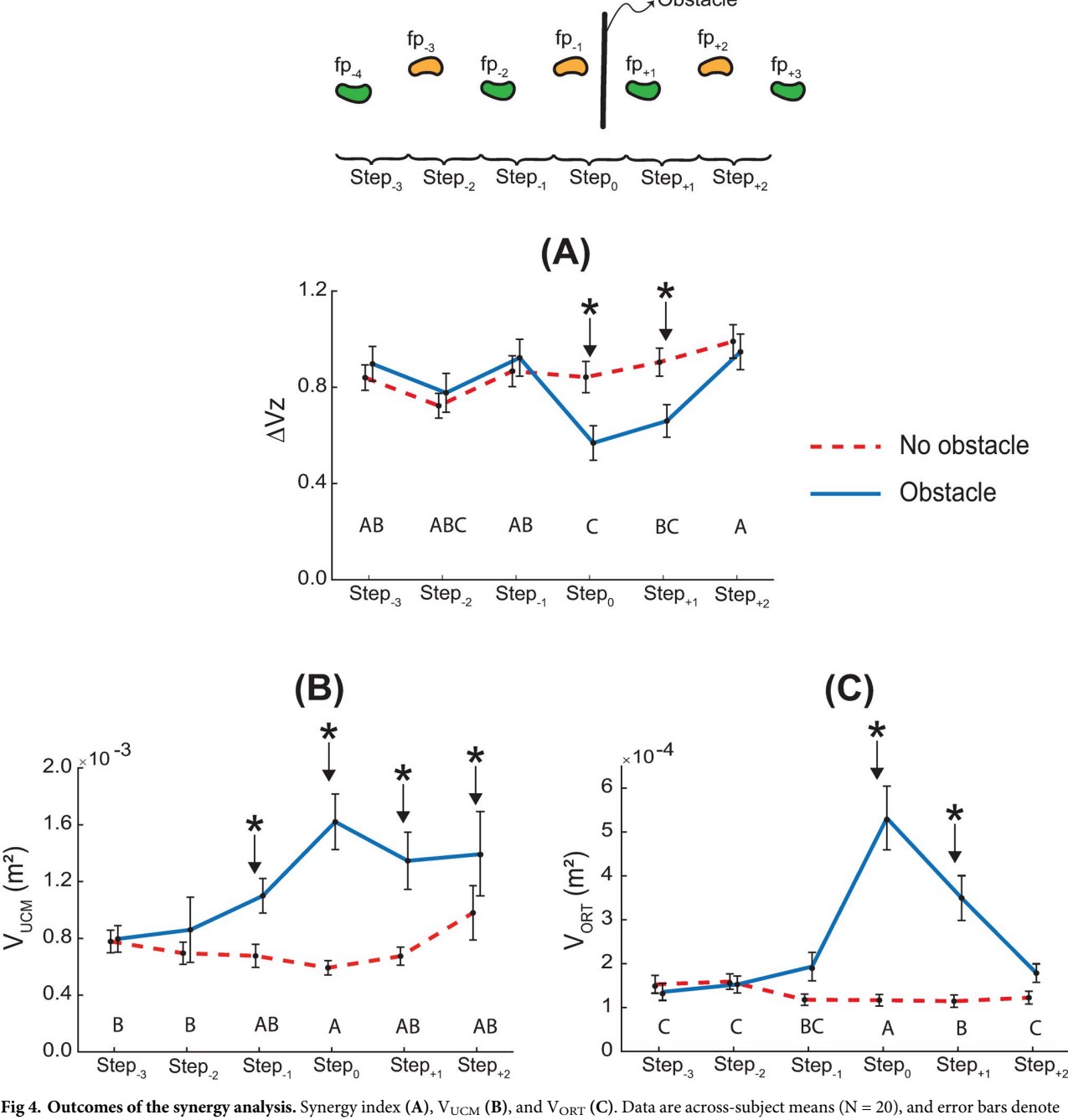

**Fig 4. Outcomes of the synergy analysis.** Synergy index (**A**), $V_{UCM}$ (**B**), and $V_{ORT}$ (**C**). Data are across-subject means (N = 20), and error bars denote standard error. Asterisks denote significant across-task differences at that step ($p < 0.05$). Across-step pairwise comparisons for the obstacle-crossing task are indicated by the row of letters above the horizontal axis for each panel. Steps that do not have letters in common are significantly different from each other (i.e., A is different from B, but A is not different from AB).

## 4. Discussion

Our goals were to establish that young adults proactively change $MOS_{AP}$ while approaching and then crossing an obstacle, and that $MOS_{AP}$ is actively controlled during unobstructed as well as obstructed walking. We had hypothesized a task by step interaction for $MOS_{AP}$ (H1).

We observed proactive changes in $MOS_{AP}$ in anticipation of a threat to stability; $MOS_{AP}$ changed across steps for unobstructed versus obstructed gait (supporting H1). Next, we had hypothesized that the synergy index will be greater than zero (H2). We report the novel finding that the central nervous system responds to changes in the body's motion with a corresponding correction in step length so that the $MOS_{AP}$ itself is approximately invariant at each heel contact (supporting H2). Finally, we had hypothesized a task by step interaction for the synergy index (H3). We observed that the synergy index was lower while crossing an obstacle compared to earlier and later steps and compared to the corresponding steps during unobstructed gait (supporting H3). We argue below that (1) the proactive changes in the $MOS_{AP}$ for the obstacle-crossing task reflect a stability-efficiency tradeoff; and (2) the positive synergy indices indicate that the passive dynamic stability is not a byproduct of another process, but that the step length is actively controlled to exploit consistently the passive dynamics to achieve either efficient or more stable gait.

## 4.1 Proactive changes in $MOS_{AP}$ and the compromise between stability and energy efficiency

The first major finding of this study is that the $MOS_{AP}$ changes while approaching the obstacle. When compared to unobstructed gait, greater stability was apparent three steps before the obstacle (Fig 3A). Changes in $MOS_{AP}$ during approach and crossing resulted from reduced speed, a more posterior CoM, and, for some steps, increased step length (Fig 3B, 3C and 3D). Previous work has demonstrated that changes in $MOS_{AP}$ and $MOS_{ML}$ were evident during the swing phase immediately prior to taking unnaturally long or quick steps [33], and in $MOS_{ML}$ one step before reaching an obstacle [34]. Here we extend this finding and demonstrate that $MOS_{AP}$ changes several steps before reaching the obstacle, indicating that the transition from unobstructed to obstructed gait occurs over several steps.

Our findings extend the argument that passive dynamic stability is modulated in response to perceived risk [6,27]. For example, the substantive increase in $MOS_{AP}$ for the lead crossing step (Figs 2 and 3A) reflects cautious gait and may be a preemptive strategy against a potential trip that could lead to a forward fall [12]. Similarly, the even-more-substantive transition back to the least stable passive dynamics (lowest $MOS_{AP}$; Fig 3A) following the trail crossing step may reflect the reduced risk of a forward fall after the trail foot has crossed the obstacle, and the exploitation of passive dynamics to propel the body forward to regain gait speed.

These fluctuations in passive dynamic stability increase the energetic cost of locomotion. The increase in $MOS_{AP}$ over approach steps, and especially the positive $MOS_{AP}$ for the crossing step, indicates that the trailing leg must push off more so that the body can rotate about and beyond the stance ankle [35]. Furthermore, the lead crossing step ($step_0$) is a slower and *longer* step (Fig 3C and 3D), which is inconsistent with the typical combination of a slower and *shorter* step during unobstructed gait which reduces energetic cost [4,36]. Therefore, our results reflect a tradeoff between stability and energy efficiency, consistent with similar arguments offered in the context of stair descent [6], and model-based optimization computations of obstacle crossing behaviors [20].

In sum, inspecting changes in $MOS_{AP}$ across tasks and steps leads to the conclusion that $MOS_{AP}$ is proactively adjusted (1) during the approach, likely to facilitate the transition from unobstructed gait to the movements required to clear the obstacle, (2) while crossing an obstacle, likely to prioritize safety over energy optimality, and (3) when resuming level gait to regain gait speed. The large fluctuations in $MOS_{AP}$ reflect a tradeoff between stability and energy efficiency.

## 4.2 Active control of MOS$_{AP}$ during unobstructed and obstructed gait

**The synergy index provides evidence for the control of MOS$_{AP}$.** The second major finding of this study is that MOS$_{AP}$ is a controlled variable for obstructed as well as unobstructed gait ($\Delta Vz > 0$ for all steps; Fig 4A). In particular, the synergy index remains significantly larger than zero, even though the input variables that define the MOS$_{AP}$ change over multiple steps for the obstacle-crossing task (Figs 2 and 3). Thus, the UCM analysis provides strong quantitative evidence that MOS$_{AP}$ is controlled.

The variance components (Fig 4B and 4C) provide information regarding the underlying processes that stabilize MOS$_{AP}$. Higher V$_{ORT}$ indicates higher variability in MOS$_{AP}$, whereas higher V$_{UCM}$ indicates greater compensatory covariance between XcoM and step length. We observed that MOS$_{AP}$ is more variable for the crossing steps (up to 205% increase in V$_{ORT}$ compared to earlier steps; Fig 4C). This likely arises from the larger muscle activations and joint moments required for the crossing steps compared to unobstructed steps [21,37]. Higher activations would increase signal-dependent noise [22], which will lead to more variable MOS$_{AP}$. However, this increase is offset by an increase in V$_{UCM}$ (up to 103% increase compared to earlier steps; Fig 4B). This compensation leads to MOS$_{AP}$ stabilization overall (Fig 4A).

**Does the synergy arise from passive mechanics, or do neural mechanisms indicate active control?.** It is unlikely that passive mechanics of the gait cycle alone can explain the positive synergy indices that we observed. Rather, both passive mechanics and active control are responsible for our data. It is indeed likely that mechanics contribute; for example, a greater push off force would result in a more anterior XcoM at the next heel contact, but it would also tend to produce longer steps [3], thereby helping to maintain MOS$_{AP}$. However, the step length is not entirely determined by a passively swinging leg. Rather, activity in the leg muscles and power at the hip and knee joints indicate active control of the forward swinging limb, which will alter the step length from what a passively swinging limb would yield [38]. Spinal stretch reflex loops have long been implicated in the control of locomotion [39,40]. Feldman et al. [41] recently elucidated locomotor control based on his theory of referent configurations which incorporates spinal reflexes. The central idea in this theory is that to intentionally change a limb's position, the nervous system modifies the parameter λ, which is the threshold muscle length at which α-motor neurons are recruited and the stretch reflex is activated. Furthermore, movements are executed by setting the time courses λ(t) for various muscles. Muscle activations arise via the stretch-reflex loop in relation to their deviations from the corresponding λ to drive the muscles (and hence the body) towards its referent configuration. It is plausible that variations in muscle lengths from their reference values, arising from variations in push off forces, will engage spinal stretch reflexes that will alter step length and contribute to the observed MOS$_{AP}$ synergy [42]. This explanation is also consistent with the view that for steady state, level gait, humans rely on spinal feedback for control in the AP direction [43,44].

The contribution of neural mechanisms to the MOS$_{AP}$ synergy may be even greater for obstructed gait, and in addition to the spinal processes implicated during unobstructed gait, supraspinal processes may contribute to the synergy. We observed large fluctuations in the MOS$_{AP}$ over the approach and crossing steps (174% increase for fp$_{+1}$ over fp$_{-1}$, and a 292% decline for fp$_{+2}$ over fp$_{+1}$; Fig 3A). Nevertheless, the synergy index, although lower (38% decline for Step$_0$ relative Step$_{-1}$; Fig 4A), remained positive. The positive synergy index despite large fluctuations in related variables over consecutive steps suggests the involvement of supraspinal mechanisms. It is generally thought that supraspinal mechanisms influence synergies. The most convincing evidence is that synergies in individuals with neurological problems (persons with Parkinson's disease, stroke survivors, etc.) are altered compared to healthy, age-

matched controls (see [31] for review). Specifically, during obstacle crossing, supra-spinal mechanisms may influence the synergy index by modulating the gain on spinal reflexes. Indeed, supraspinal centers are involved in the control of the obstacle crossing steps [21]. Spinal reflex responses in the stance leg flexors are enhanced, likely due to increased activity in the prefrontal cortex, in preparation for swinging the leg over the obstacle. Furthermore, the prefrontal cortical activity remains enhanced (compared to unobstructed walking) while swinging the leg over the obstacle [21]. It is also known that visual information about the obstacle is gathered during approach and used to alter gait characteristics [45,46]. Gathering and using visual information also implicates higher brain centers in the control of obstructed gait, supporting our view that the $MOS_{AP}$ synergy arises from spinal and supraspinal neural circuits. We note both spinal and supraspinal structures are frequently mentioned as candidate neurophysiological bases of synergies [47], although the specific mechanisms are unknown.

In summary, the synergy values that we observed could arise partially from passive body mechanics. However, active neurophysiological processes at the spinal and supra-spinal level likely contribute to this signal as well, indicating active control of $MOS_{AP}$.

Our ideas parallel previous work indicating that the MOS in the medio-lateral (ML) direction is controlled during level walking [18,48–53]. The focus of previous research on ML stability–versus AP stability–is consistent with the view that level gait requires more control along the ML direction, and minimal control along the AP direction [43]. The UCM analyses performed here show that the MOS is also controlled in the AP direction by regulating step length–not only in gait tasks that require proactive adaptations, but also during unobstructed gait.

## 4.3 Limitations

As an indicator of the stability of human gait, $MOS_{AP}$ is challenging to interpret [54]. The strict interpretation is that XcoM ahead of the BOS boundary indicates that gait cannot be stopped within a step. This would indicate instability if the task were to stop walking within a step. However, when the task is to continue walking, Hof's result is more pertinent: the same condition yields a stable gait in a passive walker by ensuring that the walker does not stall [15].

The challenge in interpreting $MOS_{AP}$ for human gait arise when extending this logic to (1) different adaptive gait tasks and (2) while considering the relation of the passive dynamic stability to the overall gait stability. First, authors have interpreted $MOS_{AP}$ depending on the task. For example, Bosse et al. consider XcoM ahead of the BOS boundary as unstable gait while descending stairs [6], whereas others (including us) assume roughly the opposite for obstacle crossing tasks [5,13]. It seems that these interpretations must be evaluated on a case-by-case basis. A better approach would be to validate the interpretations by correlating $MOS_{AP}$ characteristics with fall rates. Although this is difficult to accomplish, and many biomechanical variables used to quantify stability of human gait lack such validation [54], this is an important goal for future research.

Second, $MOS_{AP}$ captures only the passive dynamic stability of the body, and the relation of the passive stability to the overall stability of human gait is not straightforward due to the contributions of active neuromuscular processes. For example, when $MOS_{AP}$ is positive, indicating that a passive walker would stall due to insufficient energy, higher active push-off by the rear leg of a human walker could overcome this deficit. This is reflected in the data. Although the relationship $MOS_{AP} < 0$ is consistently observed in stable human gait [7–9,13,55–57], the reverse relationship ($MOS_{AP} > 0$) has also been observed in some stable human walking trials [55]. Even in our data (Fig 3A), we observe positive and negative $MOS_{AP}$ values, and there were no signs of instability in the gait overall.

Nevertheless, the accumulated evidence regarding the changes in $MOS_{AP}$ across tasks and populations suggests that the passive dynamic stability might influence the ability of the person to recover from large perturbations, given that the speed and magnitude of human neuromuscular responses are bounded. Hence passive stability is enhanced when large perturbations could occur. This effect will likely be larger in older or patient populations where maximal capacities are reduced. Therefore, we suggest that studying $MOS_{AP}$ is useful; our study of $MOS_{AP}$ has revealed information regarding locomotor control, and our findings will serve as a baseline for identifying potential locomotor issues in various populations.

Another limitation of this study is that we estimated the CoM location using four pelvis markers [26]. A rigorous whole-body model could have provided slightly different estimates of the CoM motion. However, these differences would influence $MOS_{AP}$ similarly across all conditions, and using a different method to obtain CoM kinematics would likely yield similar qualitative results and overall conclusions, especially given the large effect sizes for all our outcome variables.

It can be argued that the all the steps of the unobstructed task are similar. Therefore, the data for the unobstructed task could be collapsed across the steps and compared with the steps of the obstructed task. We performed this alternate analysis and obtained results that led to identical conclusions. However, we observed effects of gait initiation and termination in our data (cf. Fig 3C). Gait was initiated one step before the examined steps, and gait was terminated one or two steps after the trail crossing step. Therefore, collapsing the data across all steps of the unobstructed task would be suspect, and the statistical approach we used is more appropriate. Using a longer walkway–with more steps before and after the obstacle–may alter the values of our outcome variables. We do not expect that these changes will influence our key conclusions (proactive changes in and control of $MOS_{AP}$). However, effects of gait initiation and termination on $MOS_{AP}$ characteristics may be worth an independent investigation.

## 5. Conclusion

We demonstrated that the XcoM and the step length covary to maintain $MOS_{AP}$, indicating that $MOS_{AP}$ is controlled during unobstructed and obstructed gait. In conjunction with the consistently low $MOS_{AP}$ values for most of the analyzed steps, we conclude that the $MOS_{AP}$ is controlled to exploit the passive dynamics and achieve forward progression at low energetic cost. Furthermore, the value of $MOS_{AP}$ is proactively altered while approaching an obstacle, and $MOS_{AP}$ shows substantial fluctuations for the two crossing steps. These changes reflect a tradeoff between stability and energy efficiency. The changes during approach and lead crossing steps indicate increasingly cautious gait with a growing preference for stability, whereas the opposite change after the trail crossing step indicates a reversion to improving efficiency when the risk to stability is reduced. Thus, our results indicate that humans exploit the passive AP body motion to meet specific ends dictated by the locomotor task. We conclude that the UCM analysis of $MOS_{AP}$ provides new information regarding the control of stability during walking, especially for gait tasks requiring proactive adaptations, and our methods could be valuable in understanding the effects of age and pathology on gait.

## Author Contributions

**Conceptualization:** Satyajit Ambike.

**Data curation:** Ashwini Kulkarni, Chuyi Cui, Satyajit Ambike.

**Formal analysis:** Ashwini Kulkarni, Satyajit Ambike.

**Investigation:** Ashwini Kulkarni, Shirley Rietdyk, Satyajit Ambike.

**Methodology:** Ashwini Kulkarni, Chuyi Cui, Satyajit Ambike.

**Project administration:** Shirley Rietdyk, Satyajit Ambike.

**Resources:** Satyajit Ambike.

**Software:** Ashwini Kulkarni.

**Supervision:** Shirley Rietdyk, Satyajit Ambike.

**Validation:** Ashwini Kulkarni, Chuyi Cui, Shirley Rietdyk, Satyajit Ambike.

**Visualization:** Ashwini Kulkarni, Chuyi Cui, Shirley Rietdyk, Satyajit Ambike.

**Writing – original draft:** Ashwini Kulkarni, Chuyi Cui, Shirley Rietdyk, Satyajit Ambike.

**Writing – review & editing:** Ashwini Kulkarni, Chuyi Cui, Shirley Rietdyk, Satyajit Ambike.

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
