## [Decision Letter · Decision Letter 0]

23 Aug 2022

PONE-D-22-18588Humans prioritize walking efficiency or walking balance based on environmental riskPLOS ONE

Dear Dr. Ambike,

Thank you for submitting your manuscript to PLOS ONE. After careful consideration, we feel that it has merit but does not fully meet PLOS ONE’s publication criteria as it currently stands. Therefore, we invite you to submit a revised version of the manuscript that addresses the points raised during the review process.

We look forward to receiving your revised manuscript.

Kind regards,

Bernard X W Liew

Academic Editor

PLOS ONE

Journal Requirements:

2. Please change "female” or "male" to "woman” or "man" as appropriate, when used as a noun (see for instance https://apastyle.apa.org/style-grammar-guidelines/bias-free-language/gender).

Reviewers' comments:

Reviewer's Responses to Questions

**Comments to the Author**

1. Is the manuscript technically sound, and do the data support the conclusions?

Reviewer #1: Partly

Reviewer #2: Partly

2. Has the statistical analysis been performed appropriately and rigorously? 

Reviewer #1: Yes

Reviewer #2: No

3. Have the authors made all data underlying the findings in their manuscript fully available?

Reviewer #1: No

Reviewer #2: Yes

4. Is the manuscript presented in an intelligible fashion and written in standard English?

Reviewer #1: Yes

Reviewer #2: Yes

5. Review Comments to the Author

Reviewer #1: Review: PONE-D-22-18588

Title: Humans prioritize walking efficiency or walking balance based on environmental risk

This manuscript describes a study undertaken to test the test the hypothesis that humans manipulate passive anterior-posterior(AP) stability via active selection of step length to either achieve energy-efficient gait or to improve stability when balance is threatened. The authors utilize the extrapolated center of mass concept, and the UCM concept to show that the margin of stability during unobstructed gait is stabilized, and that during (and before) obstacle crossing, subjects choose a more careful strategy (which the authors argue is more energy efficient). Overall, this is a cleverly designed study, that is written up in a very readable fashion. However, I do have some suggestions and comments.

Major comments

1) Concerning terminology; most of the time, the authors talk about “stability”, but sometimes they also use the term “balance” (for instance, in the title). However, it is unclear how these concepts are different according to the authors. It may be good to define these terms (and maybe only use stability or balance?)

2) Regarding “passive anterior posterior stability”; how passive is this? Given that the authors argue that it is controlled, is it passive? I understand what the authors mean (i.e. that if you place your feet there, you can exploit passive dynamics), but is the stability really passive then?

3) The link where the data is provided is not accessible. I would also suggest to put the data in a more permanent repository, such as Zenodo, Figshare, Datadryad. All of these have the advantage that 1) they also provide a DOI (so the dataset is citable), and 2) they are searchable by major search engines (so the data is findable).

4) Regarding the margin of stability, and statements made about this on lines 163/164: This is ONLY true if there would be no collision loss at foot placement (which there of course is), and when there would be no push off (or, if collision loss and push off would be equal). See work on the Foot Placement Estimator (Millard et al., 2009, DOI: 10.1115/1.4000193) which does not have this problem (as it takes the collision into account) and Bruijn & van Dieën 2018(doi: 10.1098/rsif.2017.0816.) for a discussion on this. Hence, the authors need to be more carefull with such statements.

5) I think the whole story that this must be actively controlled (in the discussion), is well, only strong for the obstacle crossing. I think Patil et al (2019) (doi: 10.1016/j.jbiomech.2019.109375) would tend to disagree with this. I think at best, the argument can be made that active control CONTRIBUTES.

Minor comments:

1) Line 43; regarding push off adjustments; see also a recent preprint from our group; (https://doi.org/10.1101/2022.03.14.484283)

2) Line 179; the authors refer to equation 1, but that is not present?

3) I do really like the first part of the results (well done), and the figure that goes with it. However, I had some trouble figuring out which subplot was what; maybe include a small title above each subplot?

4) Discussion, lines 330-331:” with a corresponding correction in step length so that the MOSAP itself is invariant at each heel contact (supporting H1).” This is not fully true, as there is some variance. I would suggest to change to : ”with a corresponding correction in step length so that the MOSAP itself is MORE OR LESS invariant at each heel contact (supporting H1).”

5) Line 391; I think not only Dingwell argues this, but also much other work, for instance of van Leeuwen et al., Reimann et al, Rankin et al, etc etc.

Reviewer #2: The authors have used the AP margin of stability to understand whether control is energy vs stability focussed in differing environments, level vs perturbation of an obstacle. Additionally they use the UCM to determine if CoM and step length co-vary across these tasks in order maintain MOSAP. They find changes in MOSAP when crossing the obstacle, suggesting changes to improve stability as compared to energy efficient gait. Also the UCM indicated co-variation to maintain specific values of MOSap that were step dependent.

In general the paper was well written, but I found there was insufficient background and justification provided for particularly the main aim of determine whether energy or stability were maintained across different tasks. On this note previous literature has also drawn attention to the difficulty of understanding what anterior-posterior margin of stability means within a gait context, which has not been addressed within this manuscript where MoS has been the key parameter. Additionally, there are elements of the methodology which remain unclear, and potentially may bias the results. For example why are level steps separated into categories – there should be no differences in the level steps and so should be categorised within a single category. More specific comments can be found below.

Below are more specific comments on the manuscript.

Introduction

In general, I find this section overly long without cohesion to enable a reader to follow the justifications for the aims. Most specifically when setting up the aims for “energy efficient vs stability” as priorities of control this is unclearly stated and needs to be further justified and explained.

Line 38-41: please ensure you use appropriate references for statements based on prior publications

Line 43-46: As mentioned above, this is unclear. Further explanation needs to be provided to the reader on why they may be less able to resist perturbation etc… Separately on this note, past work on perturbations has suggested that can still be stable due to adaptations available such as step length/width adjustments to accommodate perturbations and prevent failure. This is an example of where your justification for the energy vs stability argument could do with re-writing to make very clear to the reader where this is coming from. Additionally line 46-47, this depends on the controller in use, or are you referring to a purely passive gait here? Please clarify.

Line 48-53: Recommend incorporating into above paragraphs and clearing up the argument to provide greater clarity to the reader on your justification

Line 56-57: not necessarily true, in unanticipated perturbations, close to dead-beat control has been shown.

Line 63: first used here, so please explain what you mean by “proactive stability”

Line 68-69: based on past literature I feel that more reasoning for why Margin of stability is the appropriate metric to use in this study? Past work (i.e. Bruijn et al 2013), has particularly highlighted that it is unclear as to what MOSAP shows and its meaning within the greater control context. In addition to this there needs to more set up to explain to the reader why negative indicated energy efficient walking – this is unclear, as only indicated unstable gait. Line 73 in line with this would maintaining MOSAP negative and remaining unstable be considered a strategy?

Line 79-82: unclear as previous hypothesis is that MOSAP will remain invariant so surely these two statements oppose one another?

Line 95: You have not clearly defined the meaning of positive and negative MOSAP values earlier in the introduction

Line 102: Explain why stabilisation would be made more difficult

Methods:

Line 113: What sample size did you determine for power? Was this achieved?

Line 115-116: What do you mean by vision normal or corrected-to-normal? Were some of the participants visually impaired?

Line 122: was the obstacle tipped then, if so how many times, and were these trials included?

Line 124: why were all participants required to lead obstacle crossing with their right leg? Some participants would have potentially found this “unnatural” as would be more comfortable leading with their left leg. This also forces them to “target” with a specific leg at a specific time and has potential biomechanical implications on the results as well. How was this worded to the participants?

Line 125: were these randomised or will there potentially be a “learnt” behaviour within the results as well?

Line 131: It is very unclear to me as to why level terrain data have been split into different categories. This generates false categorisation, particularly as each level step (unless the level terrain condition had differences), is same. These should be categorised within one grouping and not split into different group and then the analyses should reflect this as well.

Line 142: What were the selection criteria for the 15 good trials? Did all participants have 15 good trials? How do you account for speed within your analyses, as speed will affect the kinematics? Were variables normalised to account for anthropometric effects?

Line 145: Was the AP position of the heel used to denote stance initiation or also end stance? If the latter as well how did you account for the fact that in rear-foot running the heel will begin to move forwards before the end of the stance phase?

Line 156: why ankle and not foot markers? At what time point was leg length calculated at?

Line 157: Unclear as to what the average leg length was used for? Why was an average used and not the leg length in the particular trial when XCoM was being calculated?

Line 162-168: Many assumptions given here without theoretical background support. Suggest use the literature to help provide your reasoning, but also include further details to explain your justifications.

Line 202: Why not between tasks as well?

Line 208-210: Unclear as to why in one analysis participant is the random effect whilst in the other you use trials as the random effect. Please provide further details.

Results

Please make sure that within the results section you are only guiding the reader through your results, rather than describing the figures such as in lines 217-220. These details are for figure legends only. This will make your results section more cohesive and condensed. Most of 217-230 is not necessary as are more clearly presented in the lower paragraphs. As mentioned above in the methods section, my biggest concern is the separation of the level steps into categories as well – this is mis-leading and means comparisons between obstacle and level data are not true comparisons but only comparing across partial sets of level data.

Line 225: Please provide the evidence for this.

Line 251: suggest re-word to clearly show only Step 0 and +1 were different

Line 276: Recommend moving the Margin of stability section above the UCM as this follows earlier writings in the introduction etc… Suggest keeping the same order throughout the manuscript

Line 284-285: Does not appear the case in 4A based on the letters.

Line 293: This would be clearer if shown as a % between rear and front

Line 313: Not reflected in the letters used to depict significant differences

Line 314-315: recommend rewording as it is unclear

Discussion

Rather than stating elements “support H?”, it would be useful to remind the reader what each hypothesis was as currently it is difficult to follow which section answer which element

Line 334-335: Did not change for all steps? Also if level steps were considered as one category rather than randomly split, the differences would only likely occur in FP+1 and FP+2 and potentially FP-1.

Line 338: please see the introduction comments on justifying your efficiency vs stability argument

Line 357: There are also a multitude of different potential strategies that can be used in order to cross a step.

Line 358-360: It is unclear from figures 2 & 3 whether these shifts are similar in size and direction to provide this direct compensation results in MOSAP stabilisation.

Line 364-374: Please re-word this section as currently confusing. Purely passive dynamics have been shown to be sufficient to control gait on level terrain, therefore spinal feedback isn’t always necessary.

Line 376: changes are in FP+1 and +2, by your definition these are crossing steps not approach steps.

Line 380: Please further clarify how you linking the positive synergy index to supraspinal mechanism involvement.

Line 393-396: If this is the case why did you not investigate the ML direction alongside the AP?

Line 403: but this still was not positive, and difficult to interpret what greater or smaller values actually mean for the stability of gait.

Line 405: increased step length only seen in 2 steps, not across all.

Line 410: But return to “normal gait” appears to be relatively fast as by FP+3 in figure 4 all values were within the level bounds.

Line 424-427: As mentioned several times above, I find this argument has not been sufficiently justified earlier in the manscript. Currently what the results do show is a return to “normal” level mechanics within a step of the perturbation – so close to a dead-beat controller.

Line 462-463: Unclear how the methods could be valuable in understanding age and pathology – needs to be expanded if to be included.

Figures

Figure 2: Step IDs need to be included on each subplot and will make it clearer to the reader. Large X & Y axes muddle the plots – suggest label each axes without the use of the large axes which are suggesting they are set within a larger plot. Ellipse description in the legend is unclear as to what they are depicting, and also is the flatness and direct correlation to the synergy index? If so then the approximate values represented should be shown in the figure.

Figures 3, 4: Letters used to show differences between steps, very unclear. If saying A is different from B but not AB then this is confusing as the letter A is still within the letters provided! Recommend using the same letters to show difference such that e.g. FP-2 = A; FP-1 = AB and FP+2 = B, shows that -2 and -1 are different to one another and -1 and +2 are different to one another.

6. PLOS authors have the option to publish the peer review history of their article (what does this mean?). If published, this will include your full peer review and any attached files.

Reviewer #1: **Yes: **Sjoerd Bruijn

Reviewer #2: No

---

## [Author Response · Author response to Decision Letter 0]

2 Mar 2023

The response to all reviewer comments are detailed in the 'Response to reviewers' file uploaded with this revision. 

We have updated the Ethics statement in the revised manuscript as required.

---

## [Decision Letter · Decision Letter 1]

28 Mar 2023

Humans prioritize walking efficiency or walking stability based on environmental risk

PONE-D-22-18588R1

Dear Dr. Ambike,

We’re pleased to inform you that your manuscript has been judged scientifically suitable for publication and will be formally accepted for publication once it meets all outstanding technical requirements.

Kind regards,

Bernard X W Liew

Academic Editor

PLOS ONE

Additional Editor Comments (optional):

Reviewers' comments:

Reviewer's Responses to Questions

**Comments to the Author**

1. If the authors have adequately addressed your comments raised in a previous round of review and you feel that this manuscript is now acceptable for publication, you may indicate that here to bypass the “Comments to the Author” section, enter your conflict of interest statement in the “Confidential to Editor” section, and submit your "Accept" recommendation.

Reviewer #1: All comments have been addressed

Reviewer #2: All comments have been addressed

2. Is the manuscript technically sound, and do the data support the conclusions?

Reviewer #1: Yes

Reviewer #2: Yes

3. Has the statistical analysis been performed appropriately and rigorously? 

Reviewer #1: Yes

Reviewer #2: Yes

4. Have the authors made all data underlying the findings in their manuscript fully available?

Reviewer #1: Yes

Reviewer #2: Yes

5. Is the manuscript presented in an intelligible fashion and written in standard English?

Reviewer #1: Yes

Reviewer #2: Yes

6. Review Comments to the Author

Reviewer #1: The authors have done a great job in revising the manuscript. I have no further comments.

Reviewer #2: (No Response)

7. PLOS authors have the option to publish the peer review history of their article (what does this mean?). If published, this will include your full peer review and any attached files.

Reviewer #1: **Yes: **Sjoerd Bruijn

Reviewer #2: No

---

## [Editor Report · Acceptance letter]

31 Mar 2023

PONE-D-22-18588R1 

Humans prioritize walking efficiency or walking stability based on environmental risk 

Dear Dr. Ambike:

I'm pleased to inform you that your manuscript has been deemed suitable for publication in PLOS ONE. Congratulations! Your manuscript is now with our production department. 

Kind regards, 

on behalf of

Dr. Bernard X W Liew 

Academic Editor

PLOS ONE